

# Seasonal drought predictability and forecast skill in the semi-arid endorheic Heihe River basin in Northwestern China

Feng Ma [1,2], Lifeng Luo [2], Aizhong Ye [1*], Qingyun Duan [1]

[1]State Key Laboratory of Earth Surface and Ecological Resources, Faculty of Geographical Science, Beijing Normal University, Beijing 100875, China

[2]Department of Geography, Environment, and Spatial Sciences, Michigan State University, East Lansing, Michigan, USA

*Correspondence to:* Aizhong Ye (azye@bnu.edu.cn)

**Abstract.** Endorheic and arid regions around the world are suffering from serious drought problems. In this study, a drought forecasting system based on eight state-of-the-art climate models from North American Multi-Model Ensemble (NMME) and a Distributed Time-Variant Gain Hydrological Model (DTVGM) was established and assessed over the upstream and midstream of Heihe River basin (UHRB and MHRB), a typical arid endorheic basin. The 3-month Standardized Precipitation Index (SPI3) and 1-month Standardized Streamflow Index (SSI1) were used to capture meteorological and hydrological drought, and values below -1 indicate drought events. The skill of the forecasting systems was evaluated in terms of Anomaly Correlation (AC) and Brier skill score (BSS). The UHRB and MHRB showed season-dependent meteorological drought predictability and forecast skill, with higher values during winter and autumn than that during spring. For hydrological forecasts, the forecast skill in the UHRB was higher than that in MHRB. Predicting meteorological droughts more than 2 months in advance became difficult because of complex climate mechanism. However, the hydrological drought forecasts could show some skills up to 3-6 lead months due to memory of initial hydrologic conditions (ICs) during cold and dry seasons. During wet seasons, there's no skillful hydrological predictions since lead-2 month because the dominant role of meteorological forcings. During spring, the improvement of hydrological drought predictions is the most significant as more streamflow was generated by seasonal snowmelt. Besides meteorological forcings and ICs, human activities have reduced the hydrological variability and increased hydrological predictability during the wet seasons in the MHRB.





## 1    Introduction

Drought is among the most costly natural hazards in many parts of the world. It is defined as a prolonged period of

below-average rainfall, leading to water shortages in soil moisture and the hydrologic system. Drought can have a substantial

effect on many sectors, such as agriculture, ecosystem and economy, and its impacts can vary from region to region.

Mitigation of drought impact requires improved understanding of the predictability of drought and the capability to predict

drought at sufficient lead times (Below et al., 2007; Sheffield and Wood, 2012; Smith and Katz, 2013). Statistical, dynamic

and hybrid (statistical-dynamic) methods have been used for drought predictions (Mariotti et al., 2013; Hao et al., 2018;

Mishra and Singh, 2011; Pozzi et al., 2013; Luo and Wood, 2007; Luo et al., 2008). The statistical method is based on the

historical relationship between some aspects of drought and a number of predictors (e.g., large-scale climate signals). The

dynamic method relies on the skill of state-of-the-art general circulation models (GCMs) or hydrologic models that represent

physical processes linked with drought development. A hybrid method combines the statistical and dynamic methods, which

has been shown to improve drought prediction in certain case studies (Pan et al., 2013; Schepen et al., 2016). However,

drought remains one of the least understood natural hazards that are affected by many contributing factors, including

meteorological anomalies, land-atmosphere interaction and human activities (Van Loon et al., 2016a, b), which makes

accurate drought prediction a challenge (Hao et al., 2018).

Recently, climate forecasts from the North American Multi-Model Ensemble (NMME; Kirtman et al., 2014) have been

widely applied to drought predictions globally and regionally (Ma et al., 2015, 2017; Mo and Lyon, 2015; Thober and Kumar,

2015; Yuan and Wood, 2013; Yuan, 2016). The NMME-based climate forecasts (e.g., precipitation and temperature) for

hydrometeorological forecasts exhibit some improvement in skills over the reference forecasts such as climatology forecasts,

persistence forecasts or ensemble streamflow prediction (ESP) (e.g., Ma et al., 2015, 2017; Mo and Lettenmaier, 2014;

Shukla et al., 2016; Yuan, 2016). However, the improvement varies with different regions and seasons, and the understanding

of its application in endorheic and semi-arid and arid basins remains poor. Endorheic regions cover ~11.4% of global land,

which are mostly located in semi-arid and arid regions (Li et al., 2018). The semi-arid and arid regions occupy

approximately 40% of earth's land surface, and show an accelerated expansion trend (Huang et al., 2016b). The fragile



ecosystems in such regions are sensitive to climate change and human activities (Huang et al., 2016a). Therefore, the aim of this study is to fill the gaps in understanding drought predictions and predictability in endorheic and arid basins by

addressing the following questions:

(1)    How do climate forecasts perform for meteorological and hydrological drought forecasting in an endorheic river basin?

(2)    How do meteorological forcings, initial hydrologic conditions and human activities influence hydrological predictability?

Here, predictability is considered as the possible maximum forecast skill that a forecast system can achieve (Luo and Wood,

2006). The Heihe River basin, which is a typical endorheic river basin in the semi-arid and arid region of Northwestern China, is selected in this study to address the above questions. The Heihe River basin is an important part of the historic Silk Road, and an important breadbasket in Northwestern China (Zhang et al., 2015). However, the basin is subjected to serious drought problems historically and in recent decades related to climate change and intensifying human activities (Zhang et al., 2016). Therefore, it is crucial to develop a drought forecast system to promote the development of adapting strategies for

sustainable water resource and ecological management in the Heihe River basin.

The study is organized as follows: in section 2, we provide brief description of the study region and data used in this study; in section 3, we introduce the framework of the study and methods used; in section 4, we present the analysis results and discussion, followed by conclusions in section 5. Because few studies have focused on dynamic drought predictions based on GCMs and/or hydrological models in the region, this study will offer new clues for development of more accurate

drought monitoring and forecasting system.

## 2    Study area and data

### 2.1    The Heihe River basin

The Heihe River basin (HRB, Fig. 1) is the second largest inland river basin of China, located deep in the hinterland of Eurasia. The river originates from north side of Qilian Mountain with drainage area of 128,900 km$^2$ (37°41′- 42°42′N,

96°42′- 102°00′E; Ma and Frank, 2006). The HRB has an apparent landscape, ecological and climate gradient from upstream

to downstream. The landscape varies from glaciers and alpine biomes in the upstream to oases with irrigated agriculture in the midstream, to riparian ecosystems and vast Gobi desert in the downstream. Most precipitation is concentrated in the upstream during wet season (June-September). During spring and summer, as temperature rises, snow and glaciers melting and permafrost thawing generally occur. Most of streamflow in the HRB originates from precipitation, snowmelt, glaciers

melting and permafrost thawing in the upstream mountains (Wang et al., 2008), which contribute approximately 71%, 25% and 4% for the total runoff (Li et al., 2018). Most water consumption happens in the midstream for human activities (e.g., agricultural irrigation). In this study, we focus on the upstream (UHRB) and midstream (MHRB) of HRB, as these two subbasins have drastic difference in terms of impacts of human activities on hydrological processes

## 2.2   Data

Daily temperature and precipitation data at 0.5˚ spatial resolution (Zhao and Zhu, 2015) is obtained for 1961-2016, which were interpolated using thin plate spline (TPS) and 3D geospatial information from 2472 meteorological stations by the National Meteorological Information Center, China Meteorological Administration (CMA) (Hutchinson, 1998a, 1998b). Hydrological data (1982-2011) used in this study were monthly streamflow datasets from Yingluoxia (YLX) and Zhengyixia (ZYX) hydrologic stations that are located at the outlet of UHRB and MHRB. The data for hydrological model (the

Distributed Time-Variant Gain Hydrological Model (DTVGM) in this study) setup and calibration were mainly obtained from Chinese Academy of Sciences, Gansu Water Resources Bulletin, and Statistical Yearbooks, which is presented in Ma et al. (2018) in detail.

Climate hindcasts data with 1˚ ×1˚ grids (Table 1) were obtained from the North American Multi-Model Ensemble (NMME; Kirtman et al., 2014) archive (http://iridl.ldeo.columbia.edu/SOURCES/.Models/.NMME/). Monthly precipitation,

maximum, mean and minimum temperature data covering 1982-2010 were used for this study. The climate models with real time forecast were selected for drought forecasting. In this study, lead-1 month refers to forecasts initialized at the beginning of one month for itself, lead-2 is that for the next month.

## 3   Methodology



### 3.1 Meteorological and hydrological drought index

To analyze meteorological and hydrological drought, Standardized Precipitation Index (SPI, McKee et al., 1993) and

Standardized Streamflow Index (SSI, Vicente-Serrano et al., 2012) were used in this study. In the calculation, a probability

distribution of monthly precipitation or streamflow for each month was generated and standardized using empirical

Gringorten plotting position (Farahmand and AghaKouchak, 2015; Gringorten, 1963):

$$p(x_i) = \frac{i - 0.44}{n + 0.12} \tag{1}$$

where $n$ is the time span, $i$ is the position of precipitation or streamflow time series sorted from smallest to largest, $p(x_i)$ is the

corresponding empirical probability. Finally, normalization was taken to make the index comparable over time and space.

For SPI in the upstream and midstream, catchment average precipitation in the upstream and midstream were used

respectively. SSI in the upstream and midstream was calculated using streamflow at YLX (the outlet of the upstream) and

streamflow difference between ZYX (the outlet of midstream) and YLX. SPI-3 and SSI-1 were selected to characterize the

seasonal meteorological and hydrological droughts, respectively. We select SSI-1 for its good description of hydrological

drought (e.g., Barker et al., 2016; Gustard et al., 1992; Huang et al., 2017; Ma et al., 2018). Nine different lead months

forecasts are combined with observation to construct 3-months accumulated precipitation for computing SPI3. For example,

for SPI3 in August, lead month 1 uses forecast at lead month 1 (August) combined with two months observation (June to

July). Lead month 2 means the sum of forecasts at lead month 1-2 (July and August) and one month observation in June.

Lead month 3 refers to forecasts at lead month 1-3 (June to August). In this study, drought is defined when the drought index

value is below -1.

### 3.2 Seasonal drought forecasting system

In this study, meteorological drought forecasting were produced using NMME climate forecasts, and hydrological drought

forecasting makes use of a hydrological model forced by NMME climate forecasts (Figure 2). To improve the forecast skill

and drive the hydrological model, the NMME hindcasts were bias-corrected and downscaled using the "quantile mapping"

method (Wood et al., 2002). The 1-degree monthly NMME precipitation and temperature hindcasts were interpolated into

0.5-drgree with bilinear interpolation over the Heihe River basin. Then the cumulative distribution functions (CDFs) of





NMME hindcasts and observations were constructed with all years except the target year (leave-one-out), and matched to remove model forecast bias. The daily hindcasts were then generated by matching the monthly hindcasts with the daily samples from observations. Finally, the 0.5-drgree bias-corrected daily hindcasts were bilinearly interpolated into 181 sub-basins to drive the hydrological model over the HRB. The last step is only necessary in this study as our hydrological model as described below runs on sub-basin scales instead of regular lat-lon grids.

In this study, the Distributed Time-Variant Gain Hydrological Model (DTVGM; Mao et al., 2016) was used to simulate and predict streamflow. It is a distributed, catchment-based hydrological model with modules to simulate snow, runoff, streamflow routing, water-use and reservoir operation. Day Degree Factor (DDF) method is used to compute snowmelt. The runoff module is based on the water balance equation, and the routing is based on kinematic scheme (Ye et al., 2010, 2013). Three runoff components for each sub-basin, i.e., surface runoff, sub-surface runoff and base flow are computed based on precipitation, soil hydraulic parameters, and land cover parameters and sets of calibrated parameters (Ma et al., 2018; Ye et al., 2010). The human activities module, i.e., water-use and reservoir models, can be switched on or off to simulate real streamflow and naturalized streamflow. Here, water use includes irrigation water, industrial water and domestic water, which are derived from irrigation areas and irrigation quota, industrial GDP, and population distribution, respectively. The reservoir regulation rules are defined according to reservoir storage capacity and ecological flow during wet (June-September) and dry seasons. The DTVGM model has been calibrated with human activities model turned on using observed streamflow at YLX, Gaoya and ZYX stations. The DTVGM could capture the variations of streamflow well with Nash-Sutcliffe efficiency coefficient (NSE) values greater than 0.86 and 0.52 for the UHRB and MHRB during both calibration and validation periods, respectively. The detail description of DTVGM model and its calibration process can be found in Mao et al. (2016) and Ma et al. (2018). Before making forecasts, the DTVGM was continuously run for the period of 1961-1981 to spin-up the hydrological model, and continuously driven by observed meteorological forcings from 1982 to 2010 to generate the offline initial hydrological conditions (ICs) for NMME-based and ESP-type forecast experiments. In this study, ESP (Ensemble Streamflow Prediction) forecasts (see section 3.4), which were based on ensemble of historical meteorological forcings, were used as a reference hydrological forecast.



## 3.3 Forecast Verification

The meteorological and hydrological drought forecast skills at 9 different lead months were evaluated using both

deterministic and probabilistic metrics. The deterministic metric that we use is the anomaly correlation (AC; Wilks, 2011),

which is calculated as:

$$AC = \frac{\sum_{i}^{n} F'_i O'_i}{\left\langle \sum_{i}^{n} F'^2_i \cdot \sum_{i}^{n} O'^2_i \right\rangle^{1/2}} \tag{2}$$

where $F'_i$ is the anomaly for NMME hindcasts, and $O'_i$ is the anomaly for observations; for a given lead and target

month/season and member, $i$ is the target month/season, $n$ is the time span (29 years in this study).

The probabilistic metric that we use is the Brier score (BS; Wilks, 2011), which is the mean squared error of probability

forecasts, considering that the observation is $o_k=1$ if drought occurs, otherwise $o_k=0$. The BS can be defined as:

$$BS = \frac{1}{n} \sum_{k=1}^{n} (y_k - o_k)^2 \tag{3}$$

where $k$ means a numbering of the $n$ forecast-event pairs, $y_k$ is the forecast probability. The BS is negatively oriented ($0 \leqslant BS$

$\leqslant 1$), with perfect forecast exhibiting $BS=0$. The Brier skill score (BSS; Wilks, 2011) is then calculated as:

$$BSS = 1 - \frac{BS}{BS*} \tag{4}$$

where $BS*$ is the BS for the reference system. Positive BSS value indicates a better forecast, while negative value indicates a

worse forecast than the reference system.

## 3.4 Experiment design for understanding the hydrological predictability

With the calibrated DTVGM hydrological model, two experiments were carried out to distinguish the importance of initial

hydrological conditions (ICs) and meteorological forcings in the hydrological forecasting. The first experiment is the

Ensemble Streamflow Prediction (ESP), which was initialized at the beginning of each month during 1982-2010, with 28

ensemble members of 9-month meteorological forcings taken from the same period without the target year. For example, the

ESP simulation starting in January 1982 was initialized at the first day on January, and driven by 28 9-month meteorological

forcings during January-September of 1983, 1984, …, 2010. The second experiment is the so-called reverse-ESP (revESP;



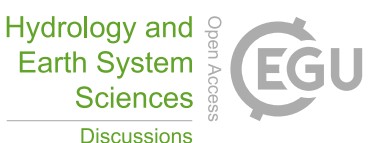

Shukla and Lettenmaier, 2011; Shukla et al., 2013; Wood and Lettenmaier, 2008), which was driven by the observed

meteorological forcings of the target year, but with 28 different ICs except the target year. For example, the revESP

simulation starting in January 1982 was driven by the meteorological forcings during January-September of 1982, but

initialized with hydrological conditions on January of 1982, 1984, …, 2010.

To determine whether the meteorological forcings or ICs are more important in the hydrologic cycle, the RMSE ratio was

calculated, which is defined as:

$$RMSE\ ratio = \frac{RMSE_{ESP}}{RMSE_{revESP}} \qquad\qquad (5)$$

where $RMSE_{ESP}$ and $RMSE_{revESP}$ are the root mean square error (RMSE) for ESP and revESP. Here, the RMSE was calculated

using the ensemble means of ESP and revESP. The RMSE ratio is lower than 1 when ICs play more important role than the

meteorological forcings in the hydrologic predictability, and the ratio is larger than 1 when the meteorological forcings

dominates.

**4    Results and discussion**

**4.1    The predictability and forecast skill for meteorological droughts**

To evaluate the performance of seasonal drought prediction system, we first examined the predictability and forecast skill of

NMME meteorological predictions based on SPI3 series in terms of AC metric (Fig. 3). The red box refers to the AC for

predictability, and the blue box is the AC for forecast skill. Here, predictability is defined by using a "perfect model"

assumption (Luo and Wood, 2006; Wang and Yuan, 2018), which considers one member of the ensemble as the

"observation", and the average of remaining members as prediction. The analysis is rotated through using all 88 ensemble

members as the "observations", and 88 values of AC are then calculated as the ensemble of predictability. For each season (3

months), there are 29-year hindcasts leading to a sample size of 87, so a correlation of 0.21 is statistically significant using a

95% significance interval.

Meteorological predictability is higher than the forecast skill in terms of AC in most case (Fig. 3), indicating some room for

improving the meteorological predictions. The meteorological predictability and forecast skill depend on the target season

because of a strong seasonality for climate in the HRB. The predictability is higher in autumn and winter than that in summer and spring, which corresponds to higher forecast skill in autumn and winter. The multi-model ensemble mean skill, shown by the blue diamonds is generally located at the upper part of the distribution, indicating that the forecast skill of grand

ensemble mean is higher than that of most members. It is not surprising that the SPI3 predictions perform well at the first two lead months, where one or two of the three months come from observations. However, as lead month increases beyond 2 months, both the predictability and forecast skill decrease significantly, where correlations of most of members are lower than 0.21.

In fact, NMME climate predictions have lower predictability and forecast skill in the northwest inland areas of China, in

comparison with Southeast monsoon regions (Ma et al., 2016). The HRB is located far from oceans and in the mid-latitude, and is little affected by sea surface temperature (SST) from oceans, especially equatorial oceans, which are the major source of predictability at seasonal time scale. Topographic influence on regional and local weather and climate cannot be resolved by GCMs, for example local ascending motion affected by Qinghai-Tibet plateau exists and have considerable impacts on precipitation over the HRB (Sun et al., 2011). In addition, the joint extreme phases of climate oscillations instead of a single

one could trigger extremes (e.g., drought) over the arid endorheic basin, and almost no climate models can capture the complex and multiple teleconnections (Shi et al., 2016).

When defining a meteorological drought as SPI3 below -1, the results of predictability and forecast skill for meteorological drought predictions are similar to that for SPI3 predictions. Figure 4 shows the BS for meteorological drought events in the upstream and midstream of Heihe River basin. AC of SPI3 reflects the forecast skill for both the dry spells and wet spells.

The BS for the different months shows the forecast skill, which primarily aims at predicting drought events. It can be seen that the NMME-based meteorological droughts show higher predictability and forecast skill at the first two lead months, especially during October-January. The results indicate that the SPI3 predictions could reasonably capture meteorological drought conditions at the first two lead months.

### 4.2 The predictability and forecast skill for hydrological droughts

Hydrological processes and predictions are more complex, especially in the MHRB. Figure 5 shows the performance of



NMME for SSI1 predictions compared to ESP, which is assessed using the AC metric. The predictions from NMME and ESP experiments are verified against DTVGM offline simulations, driven by observed meteorological forcings and calibrated against observed streamflow at Yinglouxia, Gaoya and Zhengyixia stations. Ensemble hydrological predictions from NMME show less spread than ensemble meteorological predictions, especially in the cold seasons. Due to the memory

of initial hydrologic conditions, hydrological predictions show less uncertainty than the corresponding meteorological forcings. There are notable differences in hydrological predictions between the upstream and midstream, with higher overall forecast skill in the upstream. Winter season shows the highest forecast skill, followed by autumn, spring and summer. During spring, NMME hydrological predictions show the most significant improvement over ESP, in spite of low forecast skill for precipitation predictions. During March-June, approximately 70% of streamflow is generated by seasonal snowmelt

(Wang and Li, 1999). Therefore, hydrological predictions skill may also rely more on temperature predictions , which are generally more skillful than precipitation predictions (Becker et al., 2014; Shukla et al., 2016), and the accuracy of the initial hydrological condition in terms of snow amount. In summer, NMME hydrological predictions show some improvement compared to ESP in the upstream, especially at the first lead month. However, in the midstream, low forecast skills were detected at all lead months, which are likely due to poor precipitation predictions and effects from human activities.

Predicting seasonal streamflow during summer in advance is difficult and both NMME and ESP exhibit weak skills. In autumn, lead time with good forecast skill could be up to 3 months in the upstream and 2 months in the midstream, which are similar to meteorological predictions. In addition, NMME hydrological predictions also show improvement over ESP at the first 3-4 lead months. In winter, until lead-6 month, both ESP and NMME show skillful hydrology predictions due to more important role of initial conditions, which will be discussed in the later part of this section.

To further evaluate the performance of NMME-based hydrological predictions compared with ESP for droughts (i.e., SSI1<-1), the BSS metric is used and shown in Fig. 6. The skill for hydrological drought predictions from NMME is higher than that from ESP during late spring. The improvement is even more clear for longer lead times (6-7 lead months), which may linked with the higher skill in temperature forecast. In general, NMME outperforms ESP for the 1-4 lead months (with some exceptions), but the improvement is not obvious due to long memory of initial conditions during cold season and poor

meteorological predictions during summer.



As mentioned above, the forecast skill for NMME shows notable difference between meteorological and hydrological droughts during different seasons. That is because besides meteorological forcings, initial conditions also play an important role on hydrological predictions. Figure 7 shows the relative role of initial hydrological conditions (ICs) on hydrological predictability for different months and lead times over the upstream and midstream of Heihe River basin. During cold and dry seasons (October-March), the RMSE ratio is lower than 1 at the first 2-7 months, indicating that the hydrological initial conditions play a more important role on hydrological predictability up to 2-7 lead months. The maximum lead times where the ICs prevail over the meteorological forcings in the hydrological predictability could even up to 5-7 months during October-December. In general, as the lead time increases, the contribution of initial conditions decreases while that of the meteorological forcings gradually increases over the ICs. For the forecasts in April-August, the influence of ICs could not persist for 1 month, and the meteorological forcings significantly contribute to the hydrological predictability. This means that the ICs make more contribution on hydrological predictability during cold and dry season than that during warm and wet season. This helps to explain why hydrological predictions are more skillful in dry season than that in wet season.

From the hydrological perspective, the MHRB is a human-dominant basin (Ma et al., 2018). To explore the influence of human activities on hydrological predictability, an additional experiment is conducted by turning off the human activities module in the hydrological model. The RMSE ratios of ESP and RevESP without human activities are then calculated, and the results are shown in Fig. 8. The impact of human activities is less noticeable in the upstream because less human activities are there. As for the midstream, without the impact of human activities, the RMSE ratio is less than 1 in the first lead month initialized in May and July-September (Fig. 8b). This indicates that the initial hydrological conditions have less variability in the wet seasons due to more human activities (e.g., irrigation and reservoir regulation), and human activities reduce the effect of ICs on hydrological predictability for 1 month. Therefore, the memory from ICs could only last for less than 1 month and would not contribute much to the hydrological predicting when human activities are the interference, which makes hydrological predictability rely more on meteorological prediction. In addition, the RMSE ratios in the midstream have a smaller spread for all forecast lead times when human activities are considered (Figs. 7b, 8b). This indicates that human activities reduce hydrologic variability between years, and could potentially increase the hydrological predictability.




## 5    Conclusions

Understanding the performance of climate predictions at regional or global scales provides an important basis for the utility and improvement of these products. In recent decades, drought prediction based on climate predictions at seasonal scales has improved significantly due to a range of global climate models (Hao et al., 2018). However, hydroclimatic drought prediction and predictability over an endorheic and arid basin that is affected by complex climate mechanism remains a grand challenge. A breakthrough in predicting hydroclimatic drought over endorheic basins can bring major improvements in the development of reliable drought monitoring and prediction systems at regional and global scales. In this study, the seasonal meteorological and hydrological drought predictability and forecast skill in the Heihe River basin (HRB), a typical endorheic and arid basin with distinctive characteristics from upstream to midstream, were presented in detail. The meteorological drought forecasting system was based on bias-corrected and downscaled 88-member North American Multimodel Ensemble (NMME) climate hindcasts during 1982-2010, and the hydrological drought forecasting system was established using the Distributed Time-Variant Gain Hydrological Model (DTVGM) driven by the post-processed predictions. The NMME-based hydrological predictions were compared with the ESP-type predictions, verified by offline simulations to ignoring hydrological model errors. The DTVGM with human activities modules (i.e., reservoir module and water use module) has been well calibrated over 181 subbasins in the HRB based on observed streamflow at 3 mainstream gauges and meteorological forcings for the period of 1981-2000. The Nash-Sutcliffe efficiency (NSE) at monthly scale were greater than 0.86 in the upstream during the calibration and validation periods. Given extensive irrigations and water diversions in the midstream, the NSE was greater than 0.52 indicating reasonable human activities module in the DTVGM hydrological model.

For meteorological drought predictions, the upstream and midstream show higher meteorological drought predictability than forecast skill in terms of Anomaly Correlation (AC) and Brier score (BS). The forecast skill of grand NMME ensemble mean is higher than that of most individual member. The NMME climate predictions show statistically significant predictability and forecast skills for meteorological drought in the first 2 lead months, with higher values in autumn and winter. For the hydrological drought predictions, the upstream shows higher skill than the midstream in terms of AC and Brier Skill score (BSS), due to more complex hydrological process and human activities in the midstream. The highest forecast skill occurs





during winter and autumn, while the lowest skill exists during summer. During spring, the NMME-based hydrological

predictions outperform ESP-type predictions up to 7 lead months, in spite of poor precipitation predictions. This may be due

to more seasonal snowmelt contributed to the streamflow in spring, which rely on the temperature predictions. The

NMME-based hydrological predictions show some improvement against ESP up to 2-3 months lead during autumn. The

highest forecast skill in winter could continue up to 6 months probably due to long memory of hydrological conditions.

Besides the ESP experiment, the reverse ESP was also conducted to investigate the relative role of initial conditions (ICs) on

hydrological forecasting. The role of ICs could be more significant during the cold and dry seasons, for example, ICs prevail

over the meteorological forcings up to 5-7 months during October-December, and 2-3 months during January-March.

However, the meteorological forcings outweigh the ICs at all lead months during warm and wet seasons (April-September).

That reasonably explains the season-dependent hydrological forecast skill and inconsistency between meteorology and

hydrology. In addition, a comparative experiment was conducted to explore the effect of human activities on hydrological

predictability, via removing human activities module from the DTVGM. Results show that human activities actually reduced

the hydrological variability and increased the hydrological predictability during wet seasons (May and July-September) in

the midstream. Therefore, the improvement of simulation of human activities could increase the hydrological forecast skill

over a human-dominant basin.

Although NMME-based forecasting system shows certain skill for meteorological and hydrological drought predictions,

more efforts are needed to tackle issues in the following areas: (1) understanding the physical mechanisms that caused

climate anomalies to improve climate models and the meteorological forecasting skill, as meteorological forcings play a

dominant role on hydrological predictability during wet seasons; (2) refining the human activities processes in the

hydrological forecasting system, which could facilitate the understanding of the hydrological predictions over the regions

with vast human activities; (3) improving data assimilation and observation for initial hydrological conditions (e.g., snow

and soil moisture), which could promote the development of high-precision hydrological predictions. This needs more

collaborations between scientists from different disciplines, including climate science, hydrology, agriculture, ecology and

social economy.



**Data availability.** The observed meteorological datasets, including precipitation and temperature are available at

http://data.cma.cn/data/cdcdetail/dataCode/SURF_CLI_CHN_PRE_DAY_GRID_0.5.html                                          and

http://data.cma.cn/data/cdcdetail/dataCode/SURF_CLI_CHN_TEM_DAY_GRID_0.5.html. The climate hindcasts from

NMME models are available at http://iridl.ldeo.columbia.edu/SOURCES/.Models/.NMME/.

**Author contribution.** Feng Ma, Lifeng Luo, Aizhong Ye and Qingyun Duan designed the experiments and Feng Ma carried

them out. Aizhong Ye developed the DTVGM model code and Feng Ma performed the simulations. Feng Ma prepared the

manuscript with contributions from all co-authors.

**Acknowledgements.** This work was supported by the Strategic Priority Research Program of the Chinese Academy of

Sciences (No. XDA19070104, XDA20060401), the Natural Science Foundation of China (No. 41475093), the

Intergovernmental Key International S&T Innovation Cooperation Program (No. 2016YFE0102400) and the State Key

Laboratory of Earth Surface Processes and Resource Ecology Open Research Program (NO. 2017-KF-17). This work is also

supported by the Water Science Network WaterCube Program at Michigan State University.





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





**Table 1.** Information of NMME models.

| NMME models | Spatial resolution | Hindcast | Member | Max Lead months |
|---|---|---|---|---|
| CMC1-CanCM3 | Global, 1×1 | 1981-2010 | 10 | 12 |
| CMC2-CanCM4 | Global, 1×1 | 1981-2010 | 10 | 12 |
| COLA-RSMAS-CCSM3 | Global, 1×1 | 1982-2010 | 6 | 12 |
| COLA-RSMAS-CCSM4 | Global, 1×1 | 1982-2010 | 9 | 12 |
| GFDL-CM2p1-aer04 | Global, 1×1 | 1982-2010 | 10 | 12 |
| GFDL-CM2p5-FLOR-A06 | Global, 1×1 | 1980-2010 | 12 | 12 |
| NASA-GMAO-062012 | Global, 1×1 | 1981-2010 | 7 | 9 |
| NCEP-CFSv2 | Global, 1×1 | 1982-2010 | 24 | 10 |






**Figure 1.** Geographical location of the study area (upstream and midstream of Heihe River basin) used in this study. The top

panel shows the location of the entire Heihe River basin. The bottom panel shows the geographic distribution of

hydrometeorological stations in the UHRB and MHRB.



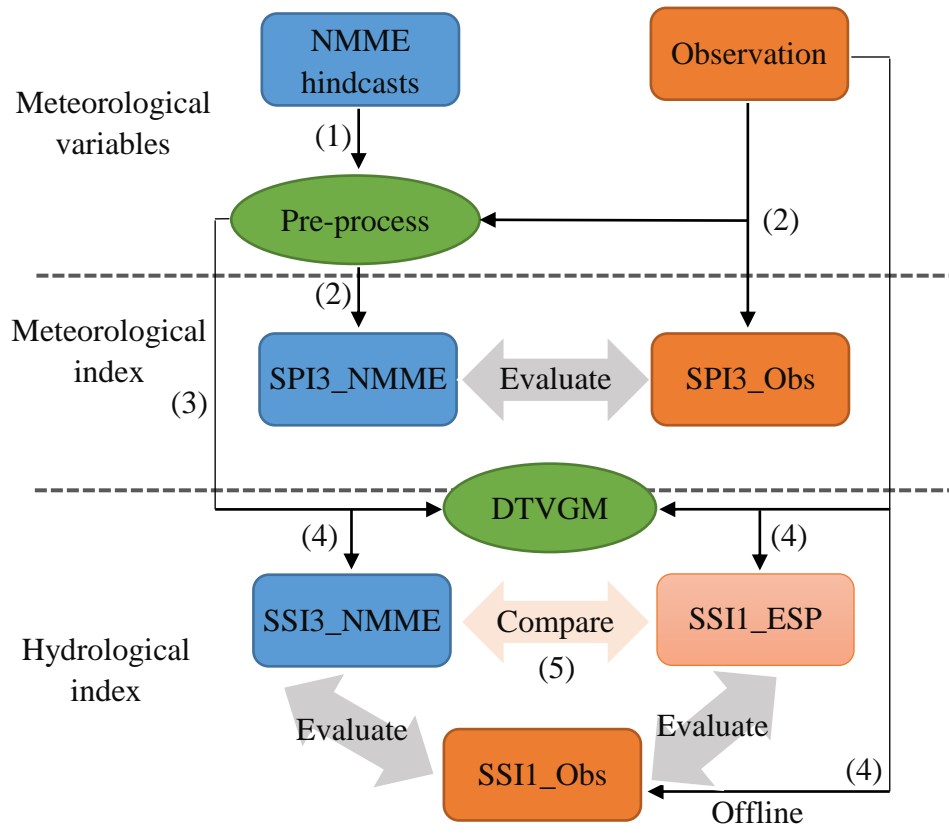

**Figure 2.** Flow chart explaining seasonal meteorological and hydrological drought forecasting system. The numbers (1)-(5)

refer to the steps for the development and assessment of a seasonal drought prediction system.



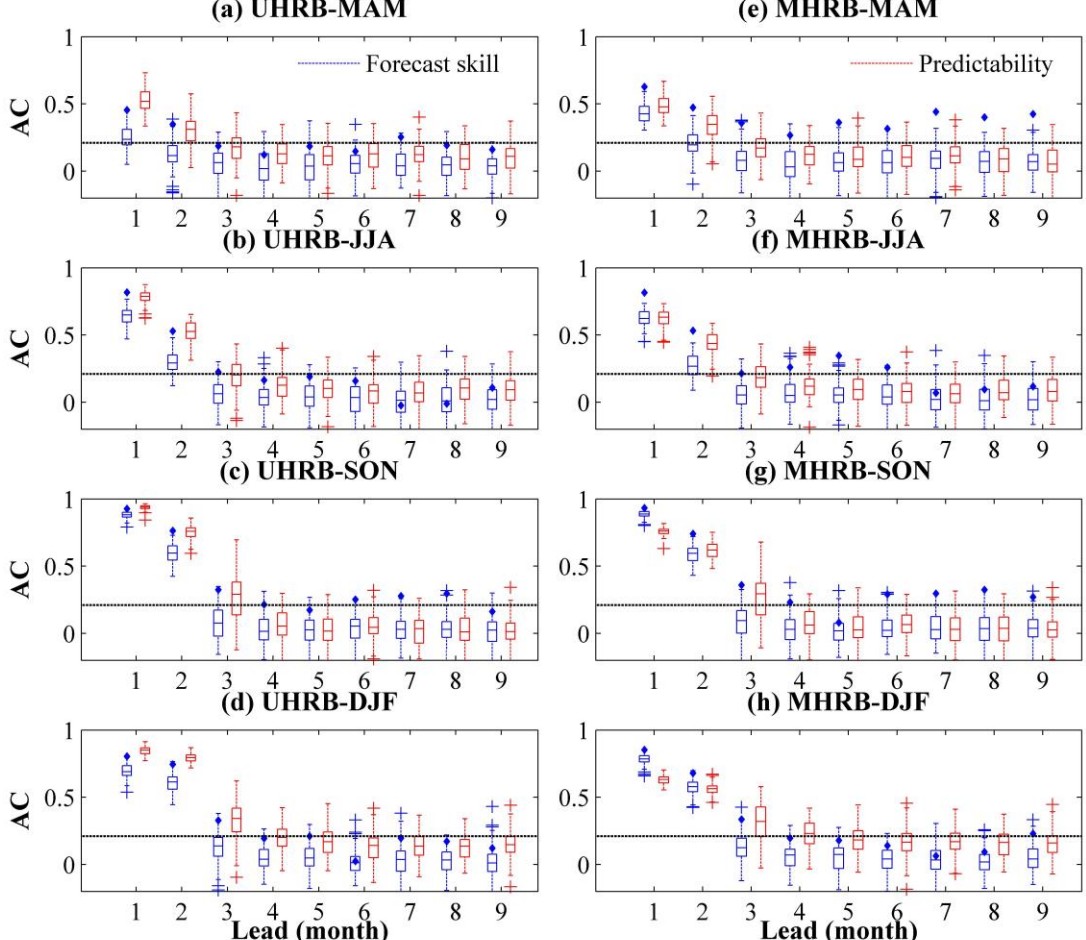

**Figure 3.** Anomaly correlation (AC) of forecast of seasonal SPI3. The red boxplots show the spread of predictability, and the blue boxplots show the spread of forecast skill for each ensemble member. The blue diamonds show the AC of the grand ensemble mean. The blue (red) crosses show the outliers for forecast skill (predictability). The dashed black line indicates the threshold (AC=0.21) of 95% confidence intervals calculated from t-test.



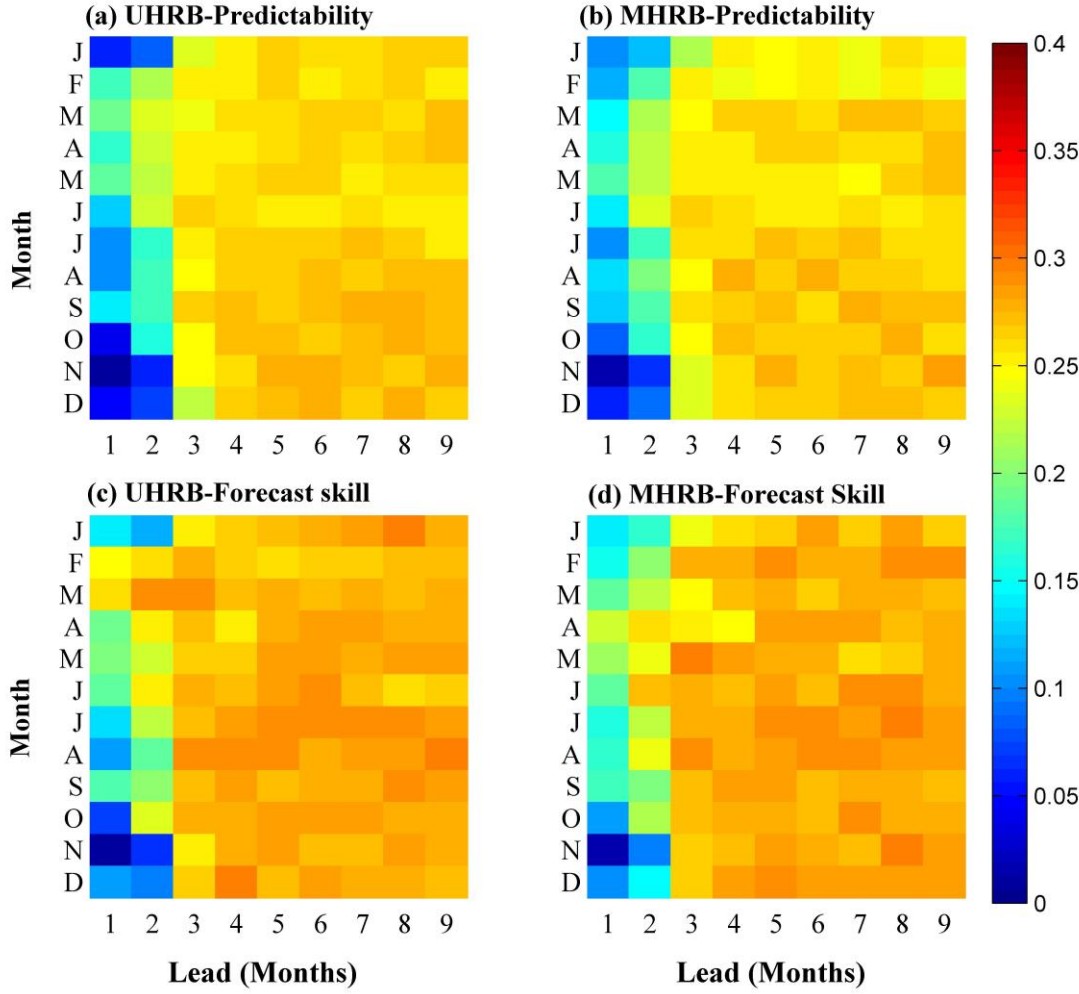

**Figure 4.** Brier score (BS) of NMME forecast for meteorological drought events. Here, a meteorological drought event happens when the SPI3 value is below -1.





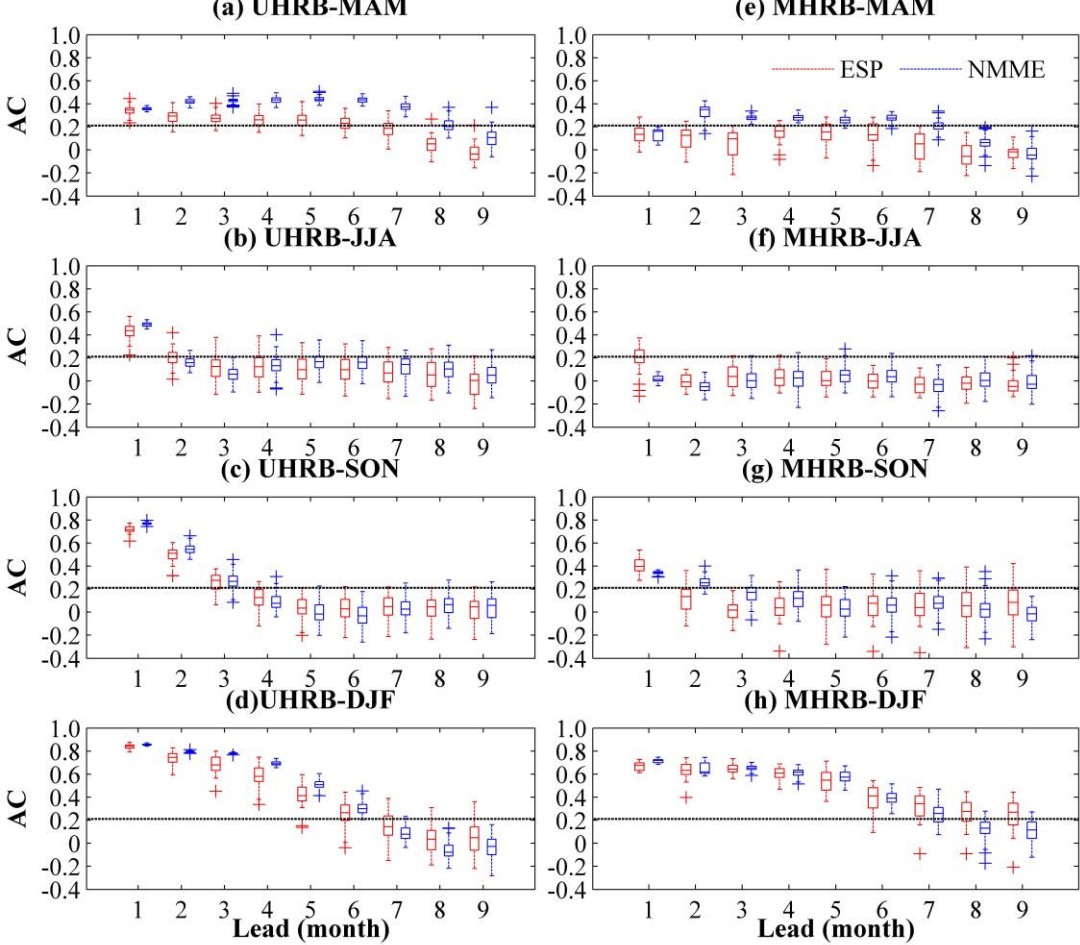


**Figure 5.** Anomaly correlation (AC) of forecast of seasonal SSI1. The red boxplots show the spread of AC of each member from NMME, and the blue boxplots show that from ESP. The blue (red) crosses show the outliers for NMME (ESP) forecast skill. The dashed black line indicates the threshold (AC=0.21) of 95% confidence intervals calculated from t-test.





**Figure 6.** Brier Skill score (BSS) of NMME forecast for hydrological drought events. Here, a hydrological drought event

happens when the SSI1 value is below -1. The reference forecasts are simulations from ESP experiment.



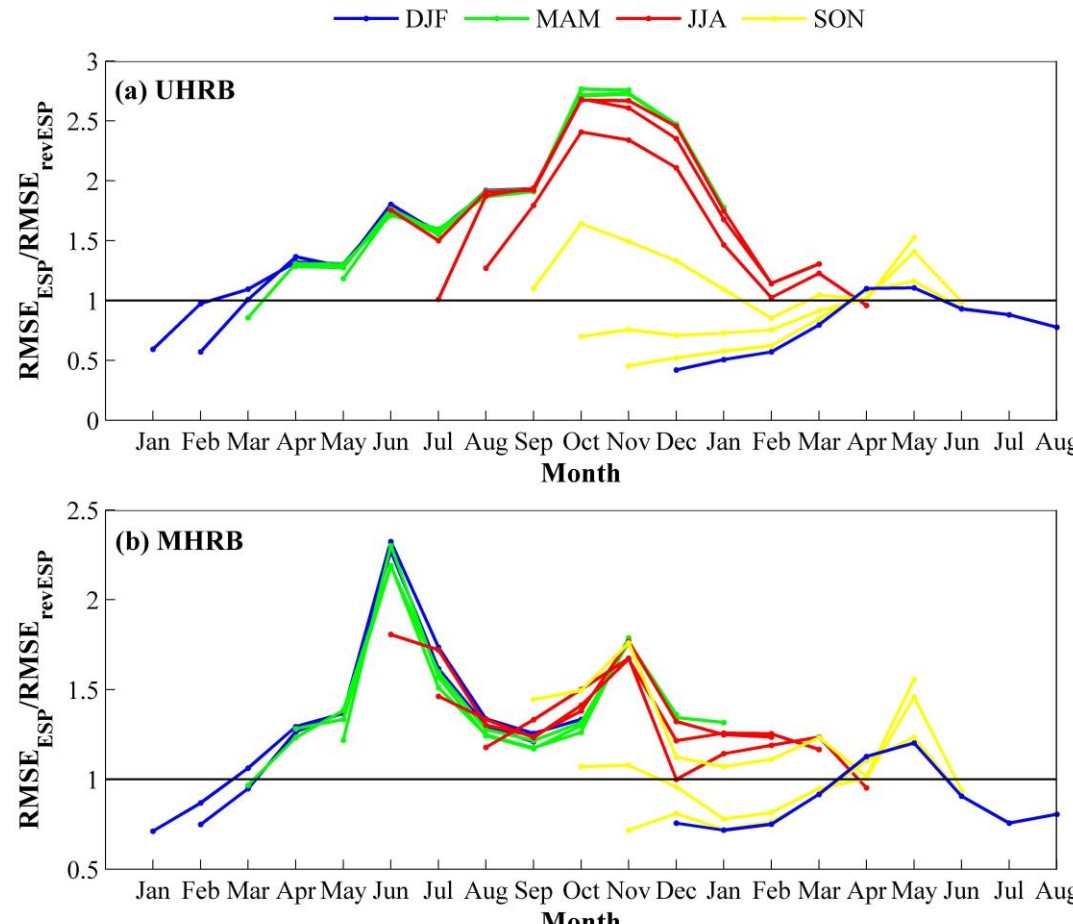

**Figure 7.** The RMSE ratio (RMSE$_{ESP}$/RMSE$_{revESP}$) as a function of start month and lead time over the upstream and midstream. The RMSE$_{ESP}$ (RMSE$_{revESP}$) is calculated between the SSI1 series from the offline simulation and that from the

ESP (revESP) experiments with human activities module switched on.



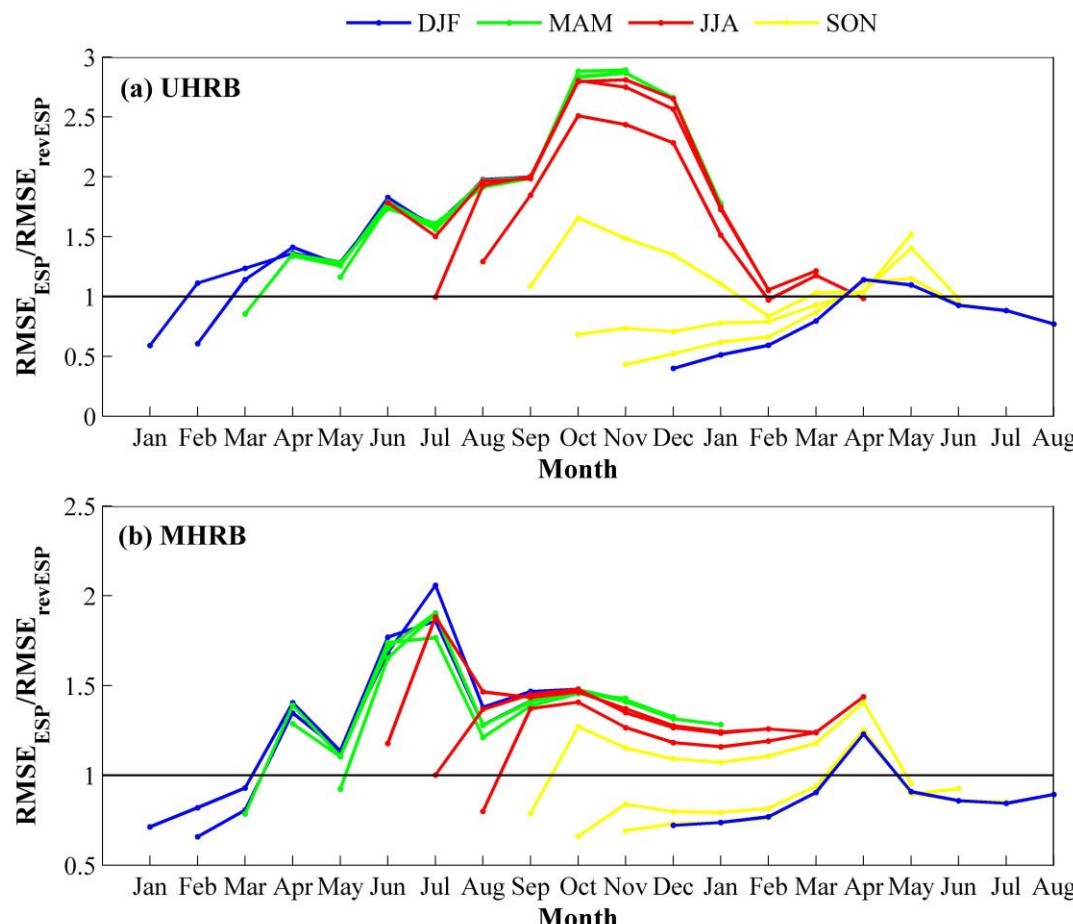

**Figure 8.** The RMSE ratio (RMSE$_{ESP}$/RMSE$_{revESP}$) as a function of start month and lead time over the upstream and midstream. The RMSE$_{ESP}$ (RMSE$_{revESP}$) is calculated between the SSI1 series from the offline simulation and that from the ESP (revESP) experiments with human activities module switched off.
