# Peer review of "Seasonal drought predictability and forecast skill in the semi-arid endorheic Heihe River basin in Northwestern China"

_Hydrology and Earth System Sciences, 2018_

## Referee Comment (RC1) · Anonymous Referee #1 · 16 Sep 2018

This study investigates seasonal drought predictability and forecast skill over a semiarid river basin. While the forecast skill evaluation is routine, the predictability is analyzed by both using a perfect model assumption and the reverse ESP framework. It is an interesting study, and the paper is easy to follow. I have a few minor comments below, basically for clarifications.

1. Given that the predictability has been quantified by both using AC with a perfect model assumption and RMSE within the reverse ESP framework, I would suggest distinguishing them in the abstract and conclusion sections. The former refers to the upper limit of forecast skill (potential skill/predictability), while the latter is usually used

for quantifying the role of initial hydrological conditions (IHCs).

2. Figures 7-8 regarding human influence on hydrological predictability is interesting. Yuan et al. (2017) also found human interventions can outweigh the climate variability for the hydrological drought forecasting over the Yellow River basin. Given that Figs. 7-8 only show the unconditional results (including dry and wet conditions) while the main focus of the paper is drought condition, a brief discussion regarding the human influence on drought predictability is encouraged.

3. Figures 5-6, are the human influence included for the reference forecast (i.e., ESP)? If so, how about the results if the human activities module is switched off?

4. A careful proofreading is necessary. I list a few typos or errors, but they may not be the complete list. L19, there's -> there are; L57, is subjected to -> is subject to; L58, intensifying -> intensified; L120, 0.5-drgree -> 0.5-degree; L233, may linked -> may be linked;

Reference: Yuan, X., M. Zhang, L. Wang, and T. Zhou, 2017: Understanding and seasonal forecasting of hydrological drought in the Anthropocene. Hydrology and Earth System Sciences, 21, 5477-5492, doi:10.5194/hess-21-5477-2017

―――――――――――――――

---

## Referee Comment (RC2) · Anonymous Referee #2 · 19 Sep 2018

The authors evaluated the predictability and forecast skill of meteorological and hydro-logical drought in the Heihe River basin based on the dynamic forecast from NMME and a hydrologic model DTVGM. The drought prediction performances for different lead time and seasons were assessed. Overall, the manuscript is well crafted with clear structures. Some grammatical errors exist and need careful proofreading. I have some minor comments.

Page 6, line 119-120: Do you use some downscaling techniques in generating the daily hindcast based on monthly data? Suggest to give some details.

Page 6, line 135: The NSE value for the MHRB is 0.52. This may lead to some uncer-

tainties in the simulated streamflow or hydrological drought and thus the performance evaluation. Suggest to mention/discuss the potential uncertainties.

Page 9, line 187-188: The authors showed that the meteorological predictability was higher in autumn and winter (than summer and spring). Any explanation/reason for this?

Figure 4: The caption is not quite informative. Suggest to give details to describe the Figure.

---

## Author Comment (AC1) · 22 Sep 2018

**View Letter**

Dear Editor and Reviewers:

Many thanks for the review comments that we received with respect to our paper. Those valuable comments have significantly enhanced our paper. We have carefully addressed the reviewers' comments and suggestions, which lead to significant revisions in many parts of the paper.

Below we hereby provide our point by point responses to the reviewer's comments.

**COMMENTS FROM EDITORS AND REVIEWERS**

**Responses to comments of Referee #1:**

This study investigates seasonal drought predictability and forecast skill over a semiarid river basin. While the forecast skill evaluation is routine, the predictability is analyzed by both using a perfect model assumption and the reverse ESP framework. It is an interesting study, and the paper is easy to follow. I have a few minor comments below, basically for clarifications.

Response: Thank you for your review and comments.

1. Given that the predictability has been quantified by both using AC with a perfect model assumption and RMSE within the reverse ESP framework, I would suggest distinguishing them in the abstract and conclusion sections. The former refers to the upper limit of forecast skill (potential skill/predictability), while the latter is usually used for quantifying the role of initial hydrological conditions (IHCs).

Response: Thank you for your suggestions. We have distinguished them in the abstract and conclusion sections.

*The predictability for meteorological drought was quantified using AC and BS with a "perfect model" assumption, referring to the upper limit of forecast skill. The hydrological predictability was to distinguish the role of initial hydrological conditions (ICs) and meteorological forcings, which was quantified by root-mean-square error (RMSE) within the ESP (Ensemble Streamflow Prediction) and reverse ESP framework.*

*Here, meteorological predictability refers to the upper limit of forecast skill using a "perfect model" assumption, while hydrological predictability is to quantify the role of initial hydrological conditions (ICs) and meteorological forcings.*

2. Figures 7-8 regarding human influence on hydrological predictability is interesting. Yuan et al. (2017) also found human interventions can outweigh the climate variability for the hydrological drought forecasting over the Yellow River basin. Given that Figs. 7-8 only show the unconditional results (including dry and wet conditions) while the main focus of the paper is drought condition, a brief discussion regarding the human influence on drought predictability is encouraged.

Response: Thank you for your suggestions. We have added the discussion regarding the human influence on drought predictability.

*Considering droughts (i.e., dry conditions), human activities could also increase hydrological drought predictability mainly by reasonable reservoir regulation. When droughts happen, discharge from reservoir plans to increase to guarantee water supply for irrigation and ecological flow, which will decrease the hydrological variability during dry periods. Therefore, human activities can outweigh meteorological variability and play a more important role on hydrological predictability. The results are similar to Yuan et al. (2017), which found human interventions can outweigh the climate variability for the hydrological drought forecasting over the Yellow River basin.*

*Yuan, X., Zhang, M., Wang, L., and Zhou, T.: Understanding and seasonal forecasting of hydrological drought in the anthropocene, Hydrol. Earth Syst. Sci., 21, 5477-5492, 2017.*

3. Figures 5-6, are the human influence included for the reference forecast (i.e., ESP)? If so, how about the results if the human activities module is switched off?

Response: Thank you for your suggestions. We have added the results when the human activities module is switched off.

*How do human activities influence hydrological drought forecast skill? Figures 9-10 show NMME-based hydrological drought forecast skill against ESP in terms of AC and BSS, when the human activities module is switched off. The forecast skill for NMME-based and ESP-based hydrological forecasts without influence of human activities (Fig. 9) are higher than that with human intervention (Fig. 5), especially in the midstream. The influence of human activities mainly occurs in the spring and early summer. Comparing Fig. 6 and Fig. 10 shows that NMME-based drought predictions have more skill improvement over the ESP-based predictions when human activities are involved. The improvement can be attained at lead times of 1-4 months in the winter, and longer lead times during April-September in the midstream. That means human activities have reduced the influence of ICs on hydrological*

*drought predictions.*

[Figure]

***Figure 5.*** *Anomaly correlation (AC) of forecast of seasonal SSI1. The red boxplots show the spread of AC of each member from NMME, and the blue boxplots show that from ESP. The blue (red) crosses show the outliers for NMME (ESP) forecast skill. The dashed black line indicates the threshold (AC=0.21) of 95% confidence intervals calculated from t-test.*

[Figure]

***Figure 6.*** *Brier Skill score (BSS) of NMME forecast for hydrological drought events. Here, a hydrological drought event happens when the SSI1 value is below -1. The reference forecasts are simulations from ESP experiment.*

[Figure]

***Figure 9.*** *Anomaly correlation (AC) of forecast of seasonal SSI1. The red boxplots show the spread of AC of each member from NMME, and the blue boxplots show that from ESP. The blue (red) crosses show the outliers for NMME (ESP) forecast skill. The dashed black line indicates the threshold (AC=0.21) of 95% confidence intervals calculated from t-test. The predictions and simulations are carried out with human activities module switched off.*

[Figure]

***Figure 10.*** *Brier Skill score (BSS) of NMME forecast for hydrological drought events. Here, a hydrological drought event happens when the SSI1 value is below -1. The reference forecasts are simulations from ESP experiment. The predictions and simulations are carried out with human activities module switched off.*

4. A careful proofreading is necessary. I list a few typos or errors, but they may not be the complete list. L19, there's -> there are; L57, is subjected to -> is subject to; L58, intensifying -> intensified; L120, 0.5-drgree -> 0.5-degree; L233, may linked -> may be linked;

Response: Thank you for your suggestions, we have checked all manuscript and corrected all errors.

*During wet seasons, there are no skillful hydrological predictions since lead-2 month because the dominant role of meteorological forcings.*

*However, the basin is subject to serious drought problems historically and in*

*recent decades related to climate change and intensified human activities (Zhang et al., 2016).*

*Finally, the 0.5-degree bias-corrected daily hindcasts were bilinearly interpolated into 181 sub-basins to drive the hydrological model over the HRB.*

*During spring, the improvement of hydrological drought predictions was the most significant as more streamflow was generated by seasonal snowmelt.*

*However, drought remains one of the least understood natural hazards that is affected by many contributing factors, including meteorological anomalies, land-atmosphere interaction and human activities (Van Loon et al., 2016a, b), which makes accurate drought prediction a challenge (Hao et al., 2018).*

*Daily temperature and precipitation data at 0.5° spatial resolution (Zhao and Zhu, 2015) are obtained for 1961-2016, which were interpolated using thin plate spline (TPS) and 3D geospatial information from 2472 meteorological stations by the National Meteorological Information Center, China Meteorological Administration (CMA) (Hutchinson, 1998a, 1998b).*

*Hydrological data (1982-2011) used in this study was monthly streamflow datasets from Yingluoxia (YLX) and Zhengyixia (ZYX) hydrologic stations that are located at the outlet of UHRB and MHRB.*

*The data for hydrological model (the Distributed Time-Variant Gain Hydrological Model (DTVGM) in this study) setup and calibration was mainly obtained from Chinese Academy of Sciences, Gansu Water Resources Bulletin, and Statistical Yearbooks, which is presented in Ma et al. (2018) in detail.*

*In this study, meteorological drought forecasting was produced using NMME climate forecasts, and hydrological drought forecasting makes use of a hydrological model forced by NMME climate forecasts (Figure 2).*

*The 1-degree monthly NMME precipitation and temperature hindcasts were interpolated into 0.5-degree with bilinear interpolation over the Heihe River basin.*

*Topographic influence on regional and local weather and climate cannot be resolved by GCMs, for example local ascending motion affected by Qinghai-Tibet plateau exists and has considerable impact on precipitation over the HRB (Sun et al., 2011).*

---

## Author Response (AR1)

[revised manuscript text omitted]

**View Letter**

Dear Editor and Reviewers:

Many thanks for the review comments that we received with respect to our paper. Those valuable comments have significantly enhanced our paper. We have carefully addressed the reviewers' comments and suggestions, which lead to significant revisions in many parts of the paper. We hope that you find the revised manuscript and the response to the reviews acceptable to HESS.

Below we hereby provide our point by point responses to the reviewer's comments.

535

**COMMENTS FROM EDITORS AND REVIEWERS**

**Responses to comments of Referee #1:**

540   This study investigates seasonal drought predictability and forecast skill over a semiarid river basin. While the forecast skill evaluation is routine, the predictability is analyzed by both using a perfect model assumption and the reverse ESP framework. It is an interesting study, and the paper is easy to follow. I have a few minor comments below, basically for clarifications.

Response: Thank you for your review and comments.

545   1.   Given that the predictability has been quantified by both using AC with a perfect model assumption and RMSE within the reverse ESP framework, I would suggest distinguishing them in the abstract and conclusion sections. The former refers to the upper limit of forecast skill (potential skill/predictability), while the latter is usually used for quantifying the role of initial hydrological conditions (IHCs).

Response: Thank you for your suggestions. We have distinguished them in the abstract and conclusion sections.

550

*The predictability for meteorological drought was quantified using AC and BS with a "perfect model" assumption, referring to the upper limit of forecast skill. The hydrological predictability was to distinguish the role of initial hydrological conditions (ICs) and meteorological forcings, which was quantified by root-mean-square error (RMSE)*

*within the ESP (Ensemble Streamflow Prediction) and reverse ESP framework.* (Line 14-18 in the marked-up manuscript version)

*Here, meteorological predictability refers to the upper limit of forecast skill using a "perfect model" assumption, while hydrological predictability is to quantify the role of initial hydrological conditions (ICs) and meteorological forcings.* (Line 292-294 in the marked-up manuscript version)

2. Figures 7-8 regarding human influence on hydrological predictability is interesting. Yuan et al. (2017) also found human interventions can outweigh the climate variability for the hydrological drought forecasting over the Yellow River basin. Given that Figs. 7-8 only show the unconditional results (including dry and wet conditions) while the main focus of the paper is drought condition, a brief discussion regarding the human influence on drought predictability is encouraged.

Response: Thank you for your suggestions. We have added the brief discussion regarding the human influence on drought predictability.

*Considering droughts (i.e., dry conditions), human activities could also increase hydrological drought predictability mainly by reasonable reservoir regulation. When droughts happen, discharge from reservoir plans to increase to guarantee water supply for irrigation and ecological flow, which will decrease the hydrological variability during dry periods. Therefore, human activities can outweigh meteorological variability and play a more important role on hydrological predictability. The results are similar to Yuan et al. (2017), which found human interventions can outweigh the climate variability for the hydrological drought forecasting over the Yellow River basin.* (Line 270-275 in the marked-up manuscript version)

*Yuan, X., Zhang, M., Wang, L., and Zhou, T.: Understanding and seasonal forecasting of hydrological drought in the anthropocene, Hydrol. Earth Syst. Sci., 21, 5477-5492, 2017.* (Line 473-474 in the marked-up manuscript version)

3. Figures 5-6, are the human influence included for the reference forecast (i.e., ESP)? If so, how about the results if the human activities module is switched off?

Response: Thank you for your suggestions. We have added the results when the human activities module is switched off.

580

*How do human activities influence hydrological drought forecast skill? Figures 9-10 show NMME-based hydrological drought forecast skill against ESP in terms of AC and BSS, when the human activities module is switched off. The forecast skill for NMME-based and ESP-based hydrological forecasts without influence of human activities (Fig. 9) are higher than that with human intervention (Fig. 5), especially in the midstream. The influence of human activities mainly*

585 *occurs in the spring and early summer. Comparing Fig. 6 and Fig. 10 shows that NMME-based drought predictions have more skill improvement over the ESP-based predictions when human activities are involved. The improvement can be attained at lead times of 1-4 months in the winter, and longer lead times during April-September in the midstream. That means human activities have reduced the influence of ICs on hydrological drought predictions.* (Line 276-283 in the marked-up manuscript version)

[Figure]

*Figure 5. Anomaly correlation (AC) of forecast of seasonal SSI1. The red boxplots show the spread of AC of each member from NMME, and the blue boxplots show that from ESP. The blue (red) crosses show the outliers for NMME (ESP) forecast skill. The dashed black line indicates the threshold (AC=0.21) of 95% confidence intervals calculated from t-test. The predictions and simulations are carried out with human activities module switched on.*

[Figure]

595

*Figure 6. Brier Skill score (BSS) of NMME forecast for hydrological drought events. Here, a hydrological drought event happens when the SSI1 value is below -1. The reference forecasts are simulations from ESP experiment. The predictions and simulations are carried out with human activities module switched on. (Line 502-511 in the marked-up manuscript*

version)

[Figure]

***Figure 9.*** *Anomaly correlation (AC) of forecast of seasonal SSI1. The red boxplots show the spread of AC of each*

*member from NMME, and the blue boxplots show that from ESP. The blue (red) crosses show the outliers for NMME*

*(ESP) forecast skill. The dashed black line indicates the threshold (AC=0.21) of 95% confidence intervals calculated*

600

*from t-test. The predictions and simulations are carried out with human activities module switched off.*

[Figure]

605

***Figure 10.*** *Brier Skill score (BSS) of NMME forecast for hydrological drought events. Here, a hydrological drought event happens when the SSI1 value is below -1. The reference forecasts are simulations from ESP experiment. The*

*predictions and simulations are carried out with human activities module switched off.* (Line 520-528 in the marked-up manuscript version)

610

4. A careful proofreading is necessary. I list a few typos or errors, but they may not be the complete list. L19, there's -> there are; L57, is subjected to -> is subject to; L58, intensifying -> intensified; L120, 0.5-drgree -> 0.5-degree; L233, may linked -> may be linked;

Response: Thank you for your suggestions, we have checked all manuscript and corrected all errors.

615

*During wet seasons, there are no skillful hydrological predictions since lead-2 month because the dominant role of meteorological forcings.* (Line 22-23 in the marked-up manuscript version)

*However, the basin is subject to serious drought problems historically and in recent decades related to climate change and intensified human activities (Zhang et al., 2016).* (Line 60-62 in the marked-up manuscript version)

620 *Finally, the 0.5-degree bias-corrected daily hindcasts were bilinearly interpolated into 140 sub-basins to drive the hydrological model over the HRB.* (Line 126-127 in the marked-up manuscript version)

*During spring, the improvement of hydrological drought predictions was the most significant as more streamflow was generated by seasonal snowmelt.* (Line 23-25 in the marked-up manuscript version)

*Daily temperature and precipitation data at 0.5° spatial resolution (Zhao and Zhu, 2015) are obtained for 1961-2016,*
625 *which were interpolated using thin plate spline (TPS) and 3D geospatial information from 2472 meteorological stations by the National Meteorological Information Center, China Meteorological Administration (CMA) (Hutchinson, 1998a, 1998b).* (Line 83-85 in the marked-up manuscript version)

*Hydrological data (1982-2011) used in this study was monthly streamflow datasets from Yingluoxia (YLX) and Zhengyixia (ZYX) hydrologic stations that are located at the outlet of UHRB and MHRB.* (Line 86-87 in the marked-up
630 manuscript version)

*The data for hydrological model (the Distributed Time-Variant Gain Hydrological Model (DTVGM) in this study) setup and calibration was mainly obtained from Chinese Academy of Sciences, Gansu Water Resources Bulletin, and*

*Statistical Yearbooks, which is presented in Ma et al. (2018) in detail. (Line 87-90 in the marked-up manuscript version)*

*In this study, meteorological drought forecasting was produced using NMME climate forecasts, and hydrological drought forecasting makes use of a hydrological model forced by NMME climate forecasts (Figure 2).* (Line 116-117 in the marked-up manuscript version)

*The 1-degree monthly NMME precipitation and temperature hindcasts were interpolated into 0.5-degree with bilinear interpolation over the Heihe River basin.* (Line 119-120 in the marked-up manuscript version)

*The RMSE ratio is lower than 1 when ICs play a more important role than the meteorological forcings in the hydrologic predictability, and the ratio is larger than 1 when the meteorological forcings dominate.* (Line 180-182 in the marked-up manuscript version)

*Topographic influence on regional and local weather and climate cannot be resolved by GCMs, for example local ascending motion affected by Qinghai-Tibet plateau exists and has considerable impact on precipitation over the HRB (Sun et al., 2011).* (Line 207-209 in the marked-up manuscript version)

**Responses to comments of Referee #2:**

The authors evaluated the predictability and forecast skill of meteorological and hydrological drought in the Heihe River basin based on the dynamic forecast from NMME and a hydrologic model DTVGM. The drought prediction performances for different lead time and seasons were assessed. Overall, the manuscript is well crafted with clear structures. Some grammatical errors exist and need careful proofreading. I have some minor comments.

Response: Thank you for your review and comments.

1. Page 6, line 119-120: Do you use some downscaling techniques in generating the daily hindcast based on monthly data? Suggest to give some details.

Response: Yes, you are right. We use the temporally downscaled techniques by sampling from the observation dataset and rescaling to match the monthly hindcasts. We have added the details in our manuscript.

*The daily hindcasts were then generated using a temporally downscaled technique by matching the monthly hindcasts with the daily samples from observations. In this approach (Luo and Wood, 2008), a randomly selected daily observation time series from the entire historical period (1961-2016) is used as a candidate for each member, and they are adjusted to match the predicted monthly values from the distributions obtained in the previous step.* (Line 122-126 in the marked-up manuscript version)

*Luo, L. and Wood, E. F.: Use of Bayesian Merging Techniques in a Multimodel Seasonal Hydrologic Ensemble Prediction System for the Eastern United States, J. Hydrometeorol., 9, 866-884, 2008.* (Line 386-387 in the marked-up manuscript version)

2. Page 6, line 135: The NSE value for the MHRB is 0.52. This may lead to some uncertainties in the simulated streamflow or hydrological drought and thus the performance evaluation. Suggest to mention/discuss the potential uncertainties.

Response: Yes, you are right. Due to vast human activities, the NSE value for the MHRB is lower, and do lead to some

670 uncertainties. We evaluated the hydrological predictions compared with offline simulations, which can reduce influence of model error but also can also lead to some uncertainties from human activities module. We have mentioned them in our manuscript.

*In the MHRB, except for input and structural errors, unrefined human activities module also increases the uncertainties*
675 *of the hydrology model, leading to the NSE value lower than that in the UHRB.* (Line 142-143 in the marked-up manuscript version)

*(2) The NSE value for the MHRB is greater than 0.52, which is still unsatisfactory. Unrefined human activities module in the hydrology model can lead to some uncertainties in the simulated streamflow and hydrological drought and thus the performance evaluation. For example, inaccurate calculations of irrigation water requirements and groundwater*
680 *can increase errors in river flow and uncertainties in the influence of human activities on hydrological droughts. Therefore, refining the human activities processes in the hydrological forecasting system, which could facilitate the understanding of the hydrological predictions over the regions with vast human activities;* (Line 329-334 in the marked-up manuscript version)

685 3. Page 9, line 187-188: The authors showed that the meteorological predictability was higher in autumn and winter (than summer and spring). Any explanation/reason for this?

Response: Thank you for your suggestions. We have added the explanation.

*Most climate anomalies (i.e., SST anomaly) occur in winter and autumn, and SST is also a potentially important*
690 *predictor (Becker et al., 2014). In addition, the climatic noise of monthly precipitation over China has obvious seasonal variation and it is greater in summer than in winter (Liu et al., 2000).* (Line 196-199 in the marked-up manuscript version)

*Becker, E., Van Den Dool, H., and Zhang, Q.: Predictability and forecast skill in NMME, J. Climate, 27, 5891-5906, 2014.* (Line 352-353 in the marked-up manuscript version)

695 *Liu, Y., Ma, K., and Lin, Z.: Potential predictability of monthly precipitation over China, J. Meteor. Res., 14, 316–329, 2000.* (Line 382 in the marked-up manuscript version)

4. Figure 4: The caption is not quite informative. Suggest to give details to describe the Figure.

Response: Thank you for your suggestions, we have added the details to describe the Figure 4.

700

*Figure 4. Brier score (BS) of NMME forecast for meteorological drought events. (a-b) Meteorological drought predictability in the upstream (a) and upstream (b); (c-d) Meteorological drought forecast skill in the upstream (c) and midstream (d). Here, a meteorological drought event happens when the SPI3 value is below -1. The BS is negatively oriented ($0 \leq BS \leq 1$), with perfect forecast exhibiting BS=0. The color from deep blue to deep red (0-0.4) means*

705 *increasing BS values, i.e., decreasing predictability or forecast skill.* (Line 497-501 in the marked-up manuscript version)